# Recent Advances in Vertically Aligned Nanowires for Photonics Applications

**DOI:** 10.3390/mi11080726

**Published:** 2020-07-26

**Authors:** Sehui Chang, Gil Ju Lee, Young Min Song

**Affiliations:** School of Electrical Engineering and Computer Science, Gwangju Institute of Science and Technology (GIST), 123 Cheomdangwagi-ro, Buk-gu, Gwangju 61005, Korea; shchangj@gm.gist.ac.kr (S.C.); gjlee0414@gist.ac.kr (G.J.L.)

**Keywords:** nanowires, photonics, LED, nanowire laser, spectral filter, coloration, artificial retina

## Abstract

Over the past few decades, nanowires have arisen as a centerpiece in various fields of application from electronics to photonics, and, recently, even in bio-devices. Vertically aligned nanowires are a particularly decent example of commercially manufacturable nanostructures with regard to its packing fraction and matured fabrication techniques, which is promising for mass-production and low fabrication cost. Here, we track recent advances in vertically aligned nanowires focused in the area of photonics applications. Begin with the core optical properties in nanowires, this review mainly highlights the photonics applications such as light-emitting diodes, lasers, spectral filters, structural coloration and artificial retina using vertically aligned nanowires with the essential fabrication methods based on top-down and bottom-up approaches. Finally, the remaining challenges will be briefly discussed to provide future directions.

## 1. Introduction

In recent years, nanowires originated from a wide variety of materials have arisen as a centerpiece for optoelectronic applications such as sensors, solar cells, optical filters, displays, light-emitting diodes and photodetectors [1,2,3,4,5,6,7,8,9,10,11,12]. Tractable but outstanding, optical features of nanowire arrays achieved by modulating its physical properties (e.g., diameter, height and pitch) allow to confine and absorb the incident light considerably, albeit its compact configuration. There are several well-organized reviews of nanowires for diverse applications; however, they cover an excessively broad range of the field, especially in photonics applications [13,14,15,16,17]. From this perspective, we narrow down the focus on recent works in the field of photonics using vertically aligned nanowires.

Vertically aligned nanowires are a particularly decent example of nanostructures for photonics applications from its matured fabrication techniques, which is auspicious to commercial manufacturing based on the ability of mass-production and cost-effectiveness. Nanowires are grown toward the vertical orientation by the commonest technique, the vapor-liquid-solid (VLS) method [18,19] but an additional refinement of its structural characteristics are done with the patterning process through diverse lithography techniques including electron-beam lithography (EBL), nanosphere lithography, nanoimprinting and photolithography [20,21,22,23].

It is of critical interest in vertical nanowires for photonics exercise to interpret and exploit its peculiar optical properties: spectral responses in a broad range of wavelengths, efficient light absorption and wavelength selectivity [24,25,26]. Light confinement phenomena in nanowires will be thoroughly addressed in the following section, which is definitely the most fundamental interpretation that penetrates overall vertical nanowire applications introduced in this review. Hence, the study of spectral resonances in nanowires based upon cylindrical optical waveguide theory paved the route to utilize vertical nanowires as a building block for spectral filters which not only can sensitively respond specific photon wavelength but also can convert light to photocurrent, expanding color gamut representation and photodetector applications (e.g., artificial retina and flexible imagers) [27,28,29,30]. Structure engineering of vertical nanowires can be conducted by additional shaping steps such as reactive ion etching (RIE), thermal oxidation and potassium hydroxide (KOH) etching, while precisely controlling the nanowire diameters, height and morphology [31,32,33].

This review pursues to summarize the current state-of-the-art studies in vertically aligned nanowires for photonics applications. Beginning with the elaboration of operating principles of vertically aligned nanowires with leaky/guided resonances from the dispersion equation in Section 2, geometrical variations and diverse fabrication techniques in vertical nanowire arrays are presented in terms of array arrangements, morphological view, top-down and bottom-up approaches and additional shaping methods in Section 3 and Section 4. We then mainly focus on the photonics of the vertical nanowires in Section 5, which is categorized as four applications: light-emitting diodes and lasers, spectral filters, coloration and artificial retina. Finally, the remaining challenges and further prospects will be briefly discussed in Section 6.

## 2. Operating Principles of Vertically Aligned Nanowires

Light confinement phenomena in nanowires is the powerful apparatus for photonics applications, which, in turn, elicits fascinating optical properties such as wavelength selectivity and resonance absorption enhancement. The underlying mechanism of these features in nanowires is bolstered by the optical excitation of leaky/guided modes. Light absorption with the correlation of wire diameters and wavelengths, which is indeed along with the incident light on the vertical nanowires coupling to particular leaky/guided waveguide modes, can be explained by assuming that the nanowire is an infinitely long semiconducting cylindrical waveguide. From the optical theory in case of the cylindrical waveguide based on Maxwell’s equations [34], the leaky/guided resonances are obtained by the dispersion equation below,
(1)$$\frac{{\left(k_{z}r\right)}^{2}m^{2}}{{\left(k_{0}r\right)}^{2}}{\left(\frac{1}{v^{2}}-\frac{1}{u^{2}}\right)}^{2}=\left[\frac{1}{u}\frac{{J^{\prime }}_{m}\left(u\right)}{J_{m}\left(u\right)}-\frac{1}{v}\frac{{H^{\prime }}_{m}\left(v\right)}{H_{m}\left(v\right)}\right]\left[\frac{n_{c}^{2}}{u}\frac{{J^{\prime }}_{m}\left(u\right)}{J_{m}\left(u\right)}-\frac{n^{2}}{v}\frac{{H^{\prime }}_{m}\left(v\right)}{H_{m}\left(v\right)}\right],$$
where $k_{z}$ represents the mode complex wavevector along the cylinder wire axis and $k_{0}=2\pi /\lambda$ (λ is the incident light wavelength in a vacuum). r is a radius of the cylinder, m is the mode order, $u=k_{c}r$, v=kr and $J_{m}$ and $H_{m}$ are the first kind of Bessel and Hankel functions, respectively. The transverse components of the mode wave vector inside and outside of the cylinder wire are $k_{c}$ and k at frequency ω which is given by,
(2a)$$k_{c}^{2}=n_{c}^{2}k_{0}^{2}-k_{z}^{2}$$
(2b)$$k^{2}=n^{2}k_{0}^{2}-k_{z}^{2},$$
where $n_{c}$ and n are indices of refraction in cylinder wire and an embedding medium. By substituting the Equation (1) in terms of $k_{z}$ and $k_{0}$, the dispersion of each mode can be easily identified. The solution of the dispersion equation is expressed by two hybrid leaky modes of HE and EH which are TM- and TE-dominant, respectively. In general, ${\mathrm{HE}}_{1m}$ transverse resonance modes excitation leads the enhanced light absorption in vertical nanowires with a notable peak in the absorption spectra. The lowest electromagnetic mode (m=1) of ${\mathrm{HE}}_{11}$ is particularly attractive in photonics applications because of no-cutoff allowing mode excitation in thin nanowires even under the subwavelength scale [35]. However, the leaky modes rise primarily in too small nanowires.

Wang and Leu simulated the light absorption of the vertical silicon (Si) nanowires by a contour map as a function of nanowire diameter and incident wavelength (Figure 1a,b) [36] and it shows a large spectral gap between ${\mathrm{HE}}_{11}$ and ${\mathrm{HE}}_{12}$ leaky modes resulting the wavelength selectivity. In addition, Figure 1c is a plot of electric field intensity, ${\left|E\left(r\right)\right|}^{2}$ at the transverse resonant wavelength in vertical Si nanowire with a diameter of 120 nm. Along with the Figure 1b, Si nanowire (d = 120 nm) shows the leaky mode resonances in the electric field intensity graph of ${\mathrm{HE}}_{11}$ and ${\mathrm{HE}}_{12}$ state at the wavelength of 670 and 400 nm, respectively, as shown in Figure 1c. Seo et al. also demonstrated the spectral tunability from the reflection spectra along the different diameters in vertical Si nanowire arrays. As shown in Figure 1d,e, the reflection dip from both simulated and measured results is shifted to the longer wavelength as the nanowire diameter increases, indicating the responses of the ${\mathrm{HE}}_{11}$ guided modes supported by each individual nanowires [37].

The vertical nanowire arrays are also influenced by the density or periodicity of wires, which trigger the Bloch mode of a 2D photonic crystal [38]. The electric field profiles by leaky/guided modes are nearly maintained both in dense and sparse vertical nanowire arrays but the photonic bandgaps are curved by the high-density. These curved photonic bandgaps work as a perturbation on the leaky/guided mode. Such perturbation can more confine the resonant electric fields within the nanowires and it can shift the resonant wavelengths to a shorter wavelength in the spectral regime. Furthermore, the proper density of vertical nanowire arrays can diffract or scatter the incident light toward the diverse directions and the dispersed lights can be reabsorbed by the adjacent nanowires. Based on this effect, a near-unity broadband absorber was implemented by using the closely-packed vertical GaAs nanowires [39].

## 3. Geometrical Variations

### 3.1. Nanowire Array Arrangements

One attractive characteristic of vertical nanowire arrays is that its optical and physical properties can be easily handled through variations in configuration, dimensions and height during the fabrication process. Also, it has generated great attention to control the optical and photonic features of nanowires by changing the array arrangements. For example, nanowire arrays can be simply categorized into a periodic or non-periodic arrangement. Figure 2a shows an example of tiling patterned nanowire arrays partitioned by its periodicity. The first and second columns on the left are scanning electron microscope (SEM) images of nanowire array with triangular and square pattern, respectively, which tend to attain highly dense array arrangement increasing packing fractions in the limited area. In periodically implemented nanowire array, despite its substantial areal packing fraction and overall optical absorption, it produces the low absorbing spots showing anisotropic angular absorption profile, which is detrimental in optoelectronic systems. Meanwhile, randomizing the nanowires can achieve optimal absorption over a particular wavelength range and wide incidence angles, which is preferred in photovoltaic applications [40].

To maximize photovoltaic absorption in vertical Si nanowire arrays, partially aperiodic nanowire arrangement was investigated by adopting an iterative random walk algorithm [41]. Nanowire supercell was randomly selected to increase the ultimate efficiency for photovoltaic applications and repeatedly arranged building partially aperiodic nanowire arrays, which constructs the periodic arrangement in a broad scale of the device. In this work, the partially aperiodic nanowire arrays showed better performance in the ultimate efficiency compared to the periodic nanowires regardless of the variations in the diameters and filling fraction. Although the random walk algorithm effectively applied in this study, further research is needed to get to the global optimum.

Another work successfully demonstrated that the emission pattern from the nanowire lasing system can be shaped by breaking its symmetry of the nanowire array arrangement [42]. Generally, emitted light pattern from III-nitride photonic crystal nanowires occasionally forms donut-like light pattern at the far-field owing to symmetry considerations, which is undesired for lasing systems. Quasi-aperiodic arrangement by aligning the non-periodic nanowire array unit cells was introduced to generate a uniformly-shaped emission beam. Meanwhile, Samsonova et al. investigated the hard X-ray emission from vertical ZnO nanowires with a different array arrangement as a target irradiated by femtosecond laser pulses [43]. Experiments showed that both ordered and disordered vertical ZnO nanowire targets enhance the fluence of the hard X-ray emission compared to the one from a flat surface target, which supports that vertical nanowire arrays can be utilized in various applications including X-ray spectroscopy.

For comparison the optical properties between the nanowire arrangements (e.g., periodicity, density and pitch) by the observation of photonic crystal modes, ordered and disordered nanowire arrays with equivalent diameters and height were fabricated by the top-down approach [44]. During the EBL process, a completely arbitrary pattern was generated by manually sifting the individual elements using software provided by EBL. By measuring the reflection of ordered and disordered nanowire arrays, the presence of Bloch photonic modes was clearly observed from the blue shift in the resonance of the ordered nanowires. In the reflection spectra of ordered nanowires having various pitches, the absorption has a propensity to decrease along the increase in lattice spacing.

### 3.2. Nanowire Morphology

Wide-range of studies in the nanowire growth focused upon shaping its morphology have been conducted based on their potential in optoelectronic applications. As shown in Figure 2b, Fonseka et al. mainly concentrated on manipulating the facets of [100] oriented vertical indium phosphide (InP) nanowires by the variations of growth parameters, while resulting different cross-sectional shapes such as square, rectangle, perfect or elongated octagon and hexagon [45]. Growth temperature variation changed the cross-sectional shapes from rectangle to octagon by transforming the side facets from four {011} to four {001} facets. In another work, the Si nanowires with elliptical cross-section were fabricated to achieve the imaging systems: non-mechanical zoom lens system and stereoscopic imaging system (Figure 2c) [46]. Nanowire lens which has two different focal lengths according to the linear polarization state enables to implement the front part of the non-mechanical zoom lens system. On the other hand, the stereoscopic imaging system was realized by another lens composed of elliptical vertical nanowire arrays while having a different optical axis for each polarization state.

Several efficient tapering methods in nanowire fabrication process have been suggested to achieve unusual optical and electronic properties such as enhancement of light absorption [47,48]. In the solar cell application, the tapered hydrogenated amorphous Si (a-Si:H) nanowire arrays exhibit a gradual reduction in effective refractive index and it presented better performance with respect to absorbance and reflection than the a-Si:H film and untapered nanowires [49]. Tapered structures also can be adopted to the water-splitting applications as a photocatalytic material [50]. In addition, Figure 2d shows tapered vertical Si nanowires grown by chemical vapor deposition (CVD) in mixed SiCl_4_/H_2_ at reduced pressure [51]. Another approach to enhance the light absorption using diameter modulation of the nanowires along its long-axis was proposed [52]. (Figure 2e) By controlling polymer passivation and Si etching periods over the Bosch-type deep reactive ion etching (DRIE) process, the diameter of Si nanowires was effectively modulated and the result demonstrated that the optical absorption efficiency of the nanowire arrays increased as the mean diameter and modulation period decreased. Besides, vertical Si nanowires with kinked geometry were fabricated from metal-assisted chemical etching (MACE) [53,54].

## 4. Fabrication Techniques

### 4.1. Top-Down Approach

A variety of top-down fabrication approaches has been explored as a promising method for large-scale production of well-aligned vertical nanowires. Bulk substrate (e.g., Si or GaAs) with the template layer on top is patterned from a mask by various lithographic techniques with high resolution patterning. Optical lithography has been widely used as an industrial standard for the patterning process to fabricate semiconductor devices. For controlling the geometries and patterns in nano-scale, numerous lithographic methods were applied to form refined nanowire arrays such as photolithography, electron-beam lithography (EBL), block-copolymer (BCP) lithography and nanosphere lithography (NSL) [55,56,57,58,59]. After the patterning step, following etching processes, dry or wet etching, eliminate materials and produce the vertically aligned nanowires on the patterned base substrate. Geometric features of vertical nanowires, including diameter, pitch, height and width, are typically controlled via both masking and etching step [60].

The overall fabrication process of vertically aligned Si nanowire array on the basis of the top-down approach is elaborated in Figure 3a [61]. Thin SiO_2_ layer is created on the top of a single crystalline Si substrate by thermal oxidation at the ambient temperature of 900 °C in the mixtures of H_2_ and O_2_ (14:8) for 90 min. In the patterning step, photolithography using a KrF scanner is conducted to fabricate the SiO_2_ nanodisks (pitch = 1.25 μm) on the SiO_2_ layer spin-coated with a positive photoresist. Then, anisotropic RIE with compound gasses of He, SF_6_ and O_2_ forms the vertical Si nanowires directly on the Si substrate by using a hard mask of SiO_2_ nanodisks. To reduce the diameter of Si nanowires, further thermal oxidation process was operated with the same condition as done in the earlier step. Finally, wet etching of hydrofluoric acid (HF) solution removes the SiO_2_ outer shells of Si nanowires, abridging its diameter until to the range of 80 to 170 nm. Nakamura et al. fabricated the Si nanowires with extremely high-aspect-ratio by the hybridization of DRIE and sacrificial oxidation, which successfully produced vertical Si nanowires with the aspect ratio of 100 [62].

### 4.2. Bottom-Up Approach

In the past few decades, there are extensive researches for synthetic nanowire strategies focused on a bottom-up approach to understand the growth of nanostructures, tune the geometrical dimensions during growth and form heterostructures. Vapor-liquid-solid (VLS) growth is one of the most widely used mechanisms based on the bottom-up approach for producing nanowires since it was initially proposed by Wagner and his colleague in 1964 from the observations of the Si whisker growth [18]. By introducing an impurity on the Si substrate, Si atom precipitation seeded from liquid alloy droplets using metallic catalysts such as Au-, Pt-, Ag-, Pd-, Cu- or Ni-Si alloy forms the growth of Si whiskers.

The growth of nanowire in the VLS method basically depends upon the nanoparticle disposition of metallic catalysts, which characterizes the nanowire diameter and spacing on the substrate. However, metallic catalysts might contaminate the nanowires and produce defects that affect the performance of the integrated device. From this perspective, catalyst-free nanowire growth techniques have been explored and Figure 3b shows one of bottom-up approach fabrication examples using selective area metal-organic chemical vapor deposition (SA-MOCVD) in the absence of the catalyst to construct vertically aligned GaAs nanowires [21]. First, a thin SiN layer is deposited by plasma-enhanced chemical vapor deposition (PECVD) as a mask for vertical GaAs nanowires. Then, polystyrene nanospheres solution is spin-coated on top of the SiN layer avoiding the multilayer formation of stacked nanospheres and then dry etching reduces the size of nanospheres to control the nanowire diameters. Metal deposition is carried out to constitute a shadow mask for the pattern transfer. Next, after the lift-off process of nanospheres, dry etching with CF_4_ plasma transfers the pattern on the substrate, followed by the remaining metal mask using wet etching with a HF solution. Finally, the vertically aligned GaAs nanowire arrays grow through the guide of a transferred pattern on the substrate by using MOCVD with controlled growth conditions. In addition, self-catalyzed GaAs nanowires can be fabricated on the Si substrate with lithography-free technique through the Ga droplet deposition directly on the Si substrate before the growth [63].

Organometal halide perovskite nanowire arrays are considered as potential materials for upcoming next generation of optoelectronic devices with high performance owing to its outstanding physical and optical properties in terms of optical absorption, direct energy bandgap, carrier mobility and carrier diffusion length [64]. Vertically aligned perovskite nanowires were fabricated inside of the anodic aluminum oxide (AAO) template by CVD based on vapor-solid-solid (VSS) method; the reaction of methyl-ammonium iodide (MAI) with solid metal (Pb or Sn) precursors in AAO template induced the growth of solid perovskite nanowires [65]. Unlike the VLS process, the VSS process generates nanowire growth at relatively low temperatures than eutectic temperature [66]. In VSS based Si nanowire fabrication with the collaboration of Au nanoparticles and AAO template, integrated nanowires had a slow growth rate and smaller but uniform diameters. Moreover, enormous vertical nanowire fabrication techniques based upon bottom-up approach including laser-assisted catalytic growth (LCG), molecular beam epitaxy (MBE), physical vapor deposition (PVD), hydrothermal (HT) method were proposed to control the growth, morphology and spacing of the nanowires [67,68,69].

### 4.3. Shaping Methods

Occasionally, additional shaping methods during the nanowire fabrication process are used to control the morphology of nanostructures. Sharpening process is typically conducted to shape nanostructures in various configurations such as needle-like tapering and truncated cone. Sharpened and truncated nanopillars can be constructed by additional etching steps from conventional Si nanopillar or cylindrical nanowire arrays. By adopting the anisotropic, isotropic RIE or both with SiO_2_ masks, the top of the Si nanopillars is etched faster than the bottom side resulting in the narrowed tip of the structures (Figure 3c,d) [70]. Besides, Cheng et al. successfully produced the needle-like Si nanowires on Si(001) substrate by introducing multiple Ag-nanoparticle catalytic etching/removal (ACER) cycle process [71]. Unlike the tapering method on Si substrate, a different tapering technique was proposed using Ti/Cr/Ti hybrid mask for sacrificial etching to create tapered nanowires on a GaAs substrate [72].

Diameter modulation of Si nanowires was achieved in a sinusoidal manner via the Bosch-type DRIE with the regulations of the etching period and the passivation [52]. After the periodic diameter modulation, thermal oxidation and wet etching process were introduced for further reduction in the diameter of the Si nanowires about subwavelength scale. Figure 3e shows another approach to control the periodicity in a single InAs nanowire using varied growth temperatures [73]. It is apparently seen that the middle part of the InAs wire has a smaller period compare to the side formed in higher growth temperature.

Furthermore, Wendisch et al. recently reported a solution-phase shaping method by the combination of MACE and KOH etching for producing the morphology-graded vertical Si nanowires [74]. MACE and KOH etching steps were sequentially conducted to yield bi-segmented Si nanowires with controlled lengths and diameters (Figure 3f), which provides another way to maximize absorbability of vertical Si nanowire arrays in specific wavelengths of light. Unlike the shaping methods based upon the additional etching or sharpening process mentioned above, microlenses were formed at the top of the vertically aligned Si nanowires by depositing the additional layers, which enhanced the overall absorption efficiency [75].

In case of metallic glass nanostructures, unlike conventional embossing techniques [76], thermoplastic drawing method was proposed to produce metallic glass nanowires with various shapes [77]. By controlling the pulling speed and temperature, physical dimensions (e.g., diameter and length) can be modulated and moreover, pre-patterned Si templates allow to make metallic glass arrays that have different nanostructures such as nanorods, nanowires and even nanotubes with the donut-like patterned Si template. As demonstrated by Hu et al., the thermoplastic drawing process also showed the possibility of metallic glass nanostructures with complex geometry, for instance, threaded shape by introducing spin in the pulling step, which is previously inaccessible with thermoplastic embossing techniques [78].

## 5. Photonics Applications of Vertical Nanowires

### 5.1. Light-Emitting Diodes (LEDs) and Lasers

The intrinsic properties, mechanically or optoelectronically, of the nanowires imply the viability of broad applications in photonics. The nanowires have been widely explored as a promising basement for constructing the light emitting source, originated from its remarkable optical characteristics as an optical cavity and a gain medium. In 2001, horizontal ZnO nanowire laser emission was firstly demonstrated by Johnson et al. [79]; however, its geometric arrangement has several limitations, including cavity loss and damages during nanowire transferring from the growth substrate. From this perspective, vertically aligned nanowires are advantageous in terms of waveguiding cavity, mode confinement and structure stability from transferring process [80,81]. Therefore, in this subsection, we will present recent works in light-emitting diodes and lasers using vertically aligned nanowires.

In the field of nanowire light-emitting diodes (LEDs) [82,83,84,85], vertical nanowires have great advantages such as light extraction from its wave guiding properties [86] and mechanical flexibility, which has been hindered in flexible LEDs production from conventional thin film structures. Wang et al. demonstrated RGB InGaN/GaN vertical nanowires LEDs on sapphire substrate monolithically [87]. They also reported the consistent redshift in photoluminescence emission spectra according to the InGaN/GaN nanowires diameter variations, indicating the further improvements in color tunability, which is along with the results from References [88,89].

Flexible LEDs are of interest in the past decades due to their wide-ranging applications, including wearable devices, flexible displays and biosensors. However, organic LEDs, traditional dominant in flexible displays, have several limitations in lifetime, external quantum efficiency and wavelength range. Recently, several research groups have been successfully fabricated the flexible LEDs based on the vertical semiconducting nanowires [90]. Figure 4a presents the flexible bi-color (blue and green) nitride nanowire LEDs [91]. By embedding the inorganic semiconducting nanowires into a flexible polymer substrate and adopting Ag nanowire electrodes, mechanical flexibility and stable electrical connection of the LEDs were realized. To avoid the unstable integration of red, green and blue LEDs typically conducted to fabricate white LEDs, Guan et al. demonstrated the flexible white LEDs based on down-conversion the blue nitride nanowire LEDs with yellow phosphors in PDMS layer [92]. The device generated a cool-white light and simultaneously maintained mechanical flexibility.

The first demonstration of the laser from a stimulated emission in ruby as an optical medium has brought prosperity in the field of photonics [93]. Today, the need for miniaturization to a nanoscale coherent light source has facilitated to instrument the nanowire lasers [94,95,96,97,98]. Semiconducting nanowires have remarkable wave guiding properties available not only to be an optical cavity but also an optical gain medium. In addition, the emission wavelength tunability also encourages the nanowires laser as an essential foundation in optoelectronic devices.

Figure 4c shows a nanopillar laser on Si substrate composed of GaAs shell and InGaAs core, which is operating at room temperature [99]. Direct growth on Si substrate denotes the possibility of the integration with nanoelectronics. The proposed optical cavity design produces unique helical propagating modes with strong light confinement even within the subwavelength nanopillars. Kim et al. also monolithically fabricated the InGaAs nanowire laser on silicon-on-insulator (SOI) [100]. Selective-area growth of nanowires with InGaAs core and InGaP shell effectively forms a nanobeam cavity and Figure 4d shows the interference patterns above a lasing threshold, which means the coherent radiation. Behzadirad et al. suggested the top-down approach fabrication techniques for high-quality GaN nanowire arrays in large area on a sapphire substrate to produce high aspect ratio GaN nanowire array lasers and showed simulated reflectivity of the ${\mathrm{HE}}_{11}$ mode as a function of the diameter of the nanowires (Figure 4e) [101].

### 5.2. Spectral Filters

The intriguing features of vertical nanowire arrays are that light absorption and reflection spectra changes according to the individual nanowire diameters, which provides opportunities to be utilized as a multispectral imaging system, optical filtering and sensing applications [102]. In Figure 5a, for example, Park and Crozier implemented a multispectral imaging system based on the multiple filtering by eight different vertical Si nanowire arrays embedded on the polydimethylsiloxane (PDMS) substrate [103]. Each group of nanowire arrays has different diameters while exhibiting total eight color differences that each filter channels (1–8 in Figure 5b inset) varies from visible to NIR wavelength range, as shown in Figure 5b, the optical microscopic image captured in transmission mode. Compared to the traditional dye-based color filter, proposed nanowire arrays have several advantages in terms of its facile fabrication with a one-step lithography process, which is cost-effective and expanded channels in the NIR range. In another work done by Park et al. unlike the previous work that did the multispectral imaging with a monochromatic image sensor, p-i-n photodetectors of vertical Si semiconducting nanowire arrays performed the filter-free color imaging, which is available due to the spectral sensitivities varying with the diameters, at the same time, converting absorbed light into photocurrents [104].

The Si nanowire array-based stackable optical filters (Si NW-SOFs) were proposed by Lee et al. [105]. Two PDMS layer-embedded nanowire arrays with different diameter and period (1st layer: 100 and 600 nm, 2nd layer: 160 and 1250 nm) having resonances in the visible and NIR regions. Interestingly, the resulting transmittance graph of the array stacked by two layers clearly shows the maintenance of both resonance characteristics of each individual layers (Figure 5c) and similar results were achieved in triple layer NW-SOFs. In addition, fluorescent imaging of bovine pulmonary artery endothelial (BPAE) cells was demonstrated to examine the integrated Si NW-SOFs and it sufficiently rejected the undesired excitation laser light. Kim et al. presented the spectral tunability and optical filtering of vertical Si nanowire arrays inside of the polymer membrane by directly stretching the constructed filter (Figure 5d) [106]. Exhibited color of the nanowire filter changes as the tensile force exerted the redshift of the spectral response, as shown in Figure 5e.

### 5.3. Coloration

On the basis of the spectral responses in nanostructures, there are extensive researches in the field of coloration by structure engineering to generate vivid colors for the various applications such as color printing, optically encoded data storage, optical sensors and color display, which can substitute conventional pigments and dye-based color production [102,107,108,109,110]. For instance, a great range of color was induced from horizontally-oriented Si nanowires by virtue of their light scattering of resonances and it observed by dark-field microscopic schemes [111]. In this subsection, the several works for expanding the spectrum of colors using vertical nanowires will be introduced depending upon the inherent spectral tunability of nanowires by controlling its physical dimensions such as diameter, period and height that has been referred consecutively throughout this review.

Fundamentally, Seo et al. demonstrated the ability of color modulation in vertical Si nanowires relying upon nanowire radii [37]. Nanowires were fabricated using EBL and ICP-RIE to build the square array with a size of 100 μm × 100 μm and period of 1 μm, total 10,000 Si nanowires for color observation convenience (Figure 6a). In the reflection spectra, the dip position of the nanowire arrays varying with radii is shifted toward the long wavelength range, which is well consistent with the simulation results from the finite difference time domain (FDTD) method. Unlike the vertical nanowires on the Si substrate in the work of Seo et al., Park et al. embedded the vertical Si nanowires inside of the polymeric (PDMS) medium as illustrated in Figure 6b, which enables the observation of transmission spectrum in nanowires and adds the flexibility of integrated device [112]. First, EBL and ICP-RIE based on top-down mechanisms produce the vertical Si nanowires on the Si substrate. Next, PDMS spin coating is conducted and the PDMS film is cured with a thickness of 50 μm. Then, the cured film is gently scraped from the Si substrate by using a razor blade. Translucent PDMS allows to measure transmission spectra of the implemented nanowires in PDMS presenting redshift of the transmission dip.

In Figure 6c, the range of structural colors was far expanded by stacking the Si nanowire arrays in PDMS membrane on the metal-insulator-metal (MIM) substrate as a bottom layer composed of Ag-SiO_2_-Ag layers [113]. According to the height of the SiO_2_ layer, the reflectance dip position in MIM is changed from yellow to cyan (Figure 6d). As we can see in Figure 6e, the complete assembly of nanowires and MIM, called transferable Si NWAs onto MIM (TSNA-MIM), maintains two reflectance dips of TSNA and MIM. Moreover, by stacking additional layers under the Si nanowire arrays, extra fine tuning of color representation is available, as shown in Figure 6f [61]. Also, the angular response in Si nanowires sensitive to the angle of incident light can be applied to the anti-counterfeiting systems (Figure 6g). We can see that the pattern in the amorphous Si (a-Si) sub-layer reflects along the viewing angle of the observer.

### 5.4. Artificial Retina

Biological eyes, especially the human eye, have several unique features in the process of visual information acquisition from focusing light to converting photo-signals to electrical signals. One of the distinctive features is the hemispherical retina, which has a wide field of view minimizing optical aberrations and photoreceptor cells with the abilities of high resolution, spectral response and signal processing [114]. To mimic the human retina, various researches were proposed in different fields [115,116,117]. Likewise, vertical nanowire arrays have been considered as an alternative way to demonstrate the artificial retina [118,119,120]. Gu et al. successfully implemented the biomimetic visual system, which has nearly perfect hemispherical retina with high-density artificial photoreceptors using perovskite nanowires inside of a curved porous aluminum oxide membrane (PAM) [121]. (Figure 7a,b) On the other hand, in Figure 7c, as a means for restoring the degraded photoreceptor cells, gold nanoparticle-decorated titania (Au-TiO_2_) nanowire arrays were implanted in blind mice without electrodes. Also, experimental results show that responses in green, blue and near UV lights indicate the restoration of light sensitivity in blind mice [122]. Photoreceptor cone cells mainly take charge of spectral responses (e.g., red, green and blue) collecting the color information [123]. For particular spectral responses to different wavelengths of light, GaAs nanowire forest using selective and sensitive photon sieve (SSPS) was recently proposed [63]. As shown in Figure 7d–f, SSPS nanowire forest is sensitive in different wavelengths such as R, G, B and near-infrared (NIR) responses with high absorbability, which can be utilized for the application of artificial photoreceptor.

## 6. Conclusions

There has been remarkable progress in nanostructure engineering that consequently promotes various applications using nanowires. The tunability of optical properties in vertical nanowires by controlling physical characteristics, including diameter, height, periodicity and pitch enables novel applications in photonics. In this review, we scrutinized recent advances in photonics applications, mainly in respect of four parts: light-emitting diodes and lasers, spectral filters, coloration and artificial retina. Wave guiding property and mechanical flexibility lead semiconducting vertical nanowires to be applicable for LEDs and lasers fabrication in the nanoscale regime. Stacking and modulating the vertical nanowire arrays allows to have a flexible and wavelength sensitive optical filter and moreover, to widen the structural color representation. The integration of artificial retina is also promising by exploiting the abilities of high-density array and high absorption in vertical nanowire arrays. However, several remaining challenges exist with some space for further improvements such as increased sensitivity of individual nanowires, connecting maneuvers between vertical nanowire photodetector and electrode and fabrication methods possible to produce large-scale, high-quality and well-organized vertical nanowire arrays for commercially available. Nevertheless, as much advances in nanowires have done over recent years, current works in vertical nanowires will facilitate to tackle the remaining challenges above.

## Figures and Tables

**Figure 1 micromachines-11-00726-f001:**
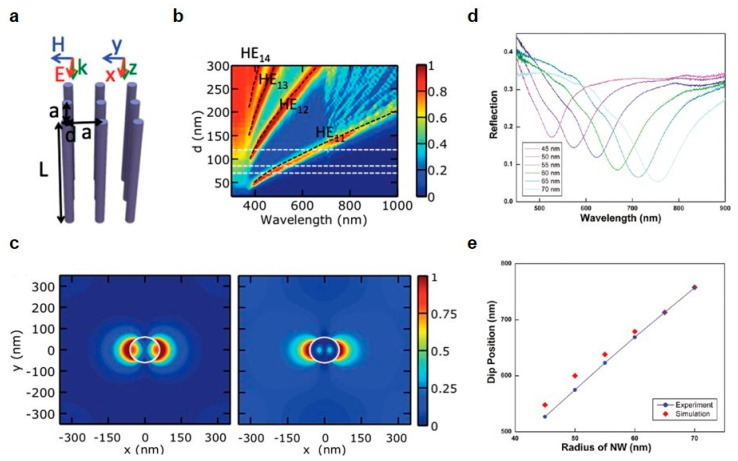
(**a**) Schematic of the vertically aligned nanowire array. (**b**) Absorption graph on the basis of wavelength and diameter of Si nanowire. (**c**) Contour maps of the electric field intensity of leaky mode resonances for vertical nanowire (d = 120 nm) at the wavelengths of (left) 670 and (right) 400 nm which accord with the ${\mathrm{HE}}_{11}$ and ${\mathrm{HE}}_{12}$ modes, respectively. Both are the top view of ${\left|E\left(r\right)\right|}^{2}$ at 0 nm in the z-axis. (**d**) Measured reflection spectra of nanowire arrays with different radii. The spectral graph shows the redshift of the spectral dip as the nanowire diameter increases. (**e**) Comparison of the measured and simulated spectral dip positions according to the changes of nanowire radius. (**a**–**c**) reprinted with permission from [36]. Copyright 2012 The Optical Society; (**d**,**e**) reprinted with permission from [37]. Copyright 2011 American Chemical Society.

**Figure 2 micromachines-11-00726-f002:**
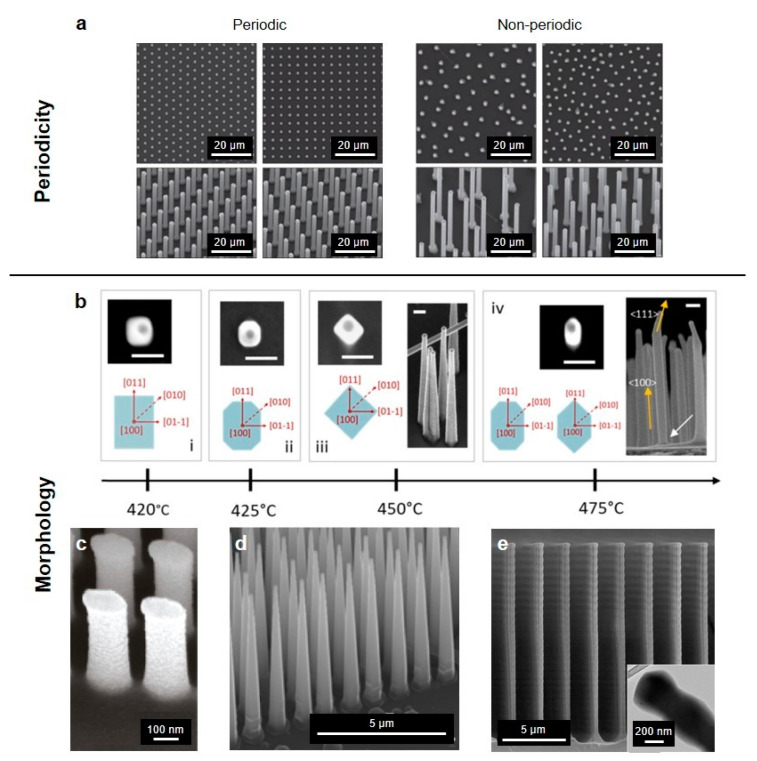
Scanning electron microscopy (SEM) images of geometrical variations in vertical nanowire arrays. (**a**) Periodic and non-periodic array arrangements. Top row: Top-view images of the nanowire arrays. Bottom row: Tilt-view (20°) images. (**b**) Side facets variations of the <100> oriented nanowires along the growth temperature. All scale bars are 100 nm. (**c**) Elliptical cross-section nanowires. The length of the semi-major and semi-minor axes are 100 nm and 50 nm, respectively. (**d**) Conical nanowires. (**e**) Diameter-modulated nanowires with the transmission electron microscopy (TEM) inset image. (**a**) reprinted by permission from Springer Nature: Springer *Nature Materials* [40], Copyright 2010; (**b**) reprinted by permission from Springer Nature: Springer *Nanoscale Research Letters* [45], Copyright 2019; (**c**) reprinted with permission from [46]. Copyright 2011 American Chemical Society; (**d**) reprinted with permission from [51]. Copyright 2011 American Chemical Society; (**e**) reprinted from [52], with permission from Wiley.

**Figure 3 micromachines-11-00726-f003:**
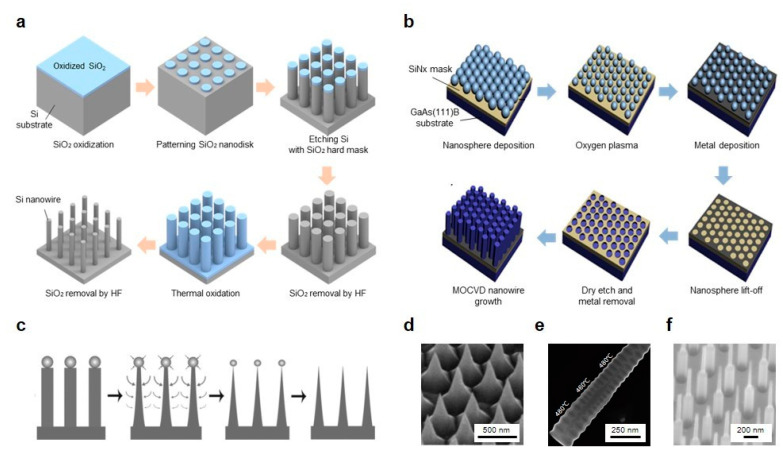
(**a**) Top-down approach of Si nanowire array fabrication. (**b**) Bottom-up approach of GaAs nanowire array fabrication. (**c**) Tapering process by controlling the etching rate. (**d**) SEM image of tapered nanowire arrays after the additional etching steps. (**e**) SEM image of a single nanowire with varying periodicity by temperature adjustment. (**f**) SEM image of bi-segmented nanowires with the (top/bottom) diameters (60/143 nm). (**a**) reprinted by permission from Springer Nature: Springer *Scientific Reports* [61], Copyright 2019; (**b**) reprinted with permission from [21]. Copyright 2012 American Chemical Society; (**c**,**d**) reprinted from [70], with the permission of AIP Publishing; (**e**) reprinted by permission from Springer Nature: Springer *Nature Nanotechnology* [73], Copyright 2008; (**f**) reprinted with permission from [74]. Copyright 2020 American Chemical Society.

**Figure 4 micromachines-11-00726-f004:**
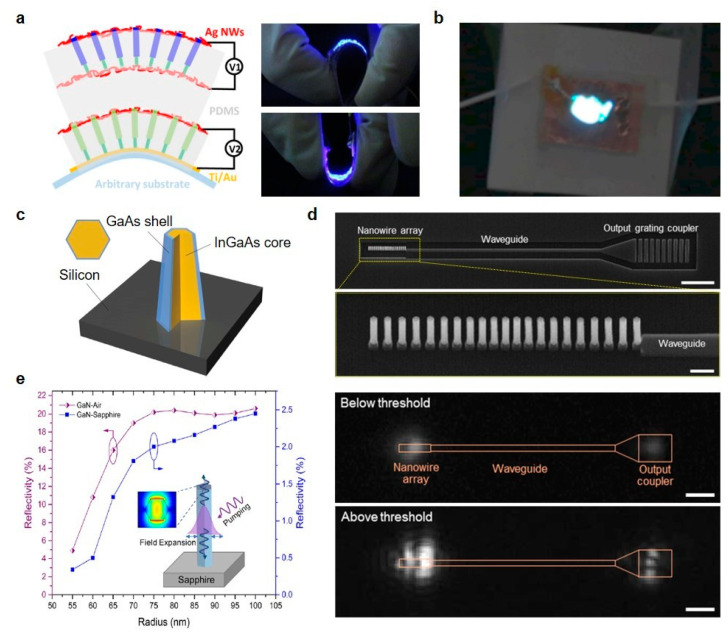
(**a**) Left: Schematic of the flexible nanowire LEDs. Right: Photographs of the mechanical flexibility of the vertical nanowire LEDs. (**b**) Photograph of the flexible white nanowire LEDs. (**c**) Schematic of the nanolaser integrated on the Si substrate. GaAs shell and InGaAs core are illustrated. (**d**) First row: SEM image of InGaAs nanowire array laser on SOI (Scale bar: 5 μm). Second row: Close-up view SEM image of the nanowire array in the first row (Scale bar: 500 nm). Third, fourth rows: Emission patterns of the nanowire array laser are presented (Scale bar: 5 μm). Interference patterns above the lasing threshold. (**e**) Simulated reflectivity of the ${\mathrm{HE}}_{11}$ mode on GaN-Air and GaN-sapphire interfaces. (**a**) Reprinted with permission from [91]. Copyright 2015 American Chemical Society; (**b**) reprinted with permission from [92]. Copyright 2016 American Chemical Society; (**c**) reprinted by permission from Springer Nature: Springer *Nature photonics* [99], Copyright 2011; (**d**) reprinted with permission from [100]. Copyright 2017 American Chemical Society; (**e**) reprinted with permission from [101]. Copyright 2018 American Chemical Society.

**Figure 5 micromachines-11-00726-f005:**
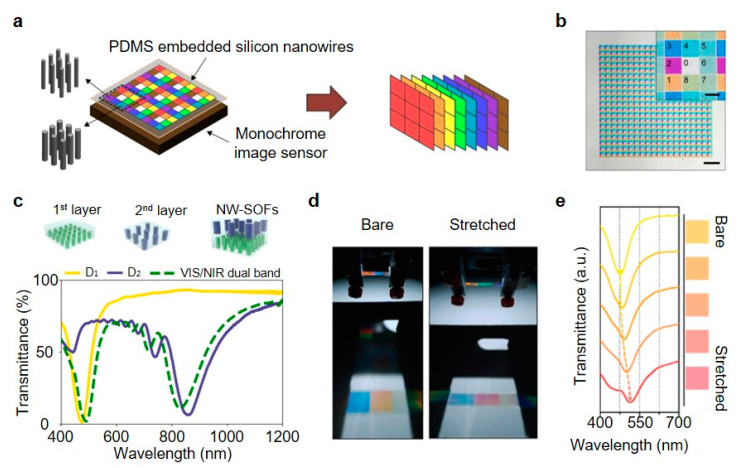
(**a**) Schematic of the multispectral filtering system with vertical Si nanowire array. (**b**) Optical microscopic image of integrated vertical Si nanowires on transparent polydimethylsiloxane (PDMS) substrate. Scale bars are 200 μm and 20 μm (inset). (**c**) Measured transmission spectra of the individual and (dotted-line) fabricated Si nanowire array-based stackable optical filters (Si NW-SOFs). The first layer has the diameter (D1) and period (Λ1) of 100 and 600 nm, respectively and 160 nm-diameter (D2) and a 1250 nm-period (Λ2) in the second layer. All are the height of 2 μm. (**d**) Photographs of the mechanotunable nanowire optical filter. (**e**) Transmittance spectra of the Si nanowire arrays and color variations of the Si nanowire arrays along the stretched degree. (**a**,**b**) reprinted by permission from Springer Nature: Springer *Scientific Reports* [103], Copyright 2013; (**c**) reprinted from [105], with permission from Wiley; (**d**,**e**) reprinted from [106], with permission from De Gruyter.

**Figure 6 micromachines-11-00726-f006:**
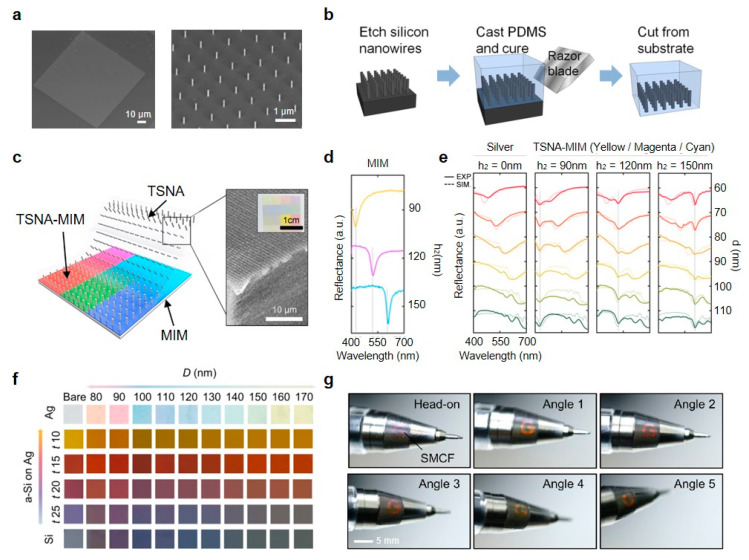
(**a**) SEM images of the square array of vertical nanowires. Top-view (left) and 30° of tilt-view (right). (**b**) Schematic of the transferring process of Si nanowires into the PDMS substrate. (**c**) Schematic of transferable Si NWAs onto MIM (TSNA-MIM), TSNA and MIM configurations. (**d**) Measured reflectance spectra for MIM for yellow, magenta and cyan colors, respectively. (**e**) Reflectance spectra of TSNA on colored MIMs. (**f**) The color table with the photographs of the integrated spectral mixing color filter (SMCF) according to the thickness and diameter of a-Si and Si nanowire array, respectively. (**g**) Photographs of SMCF-based anti-counterfeiting sticker with a variety of viewing angles. (**a**) Reprinted with permission from [37]. Copyright 2011 American Chemical Society; (**b**) reprinted from [112], with the permission of AIP Publishing; (**c**–**e**) reprinted with permission from [113]. Copyright 2019 American Chemical Society; (**f**,**g**) reprinted by permission from Springer Nature: Springer *Scientific Reports* [61], Copyright 2019.

**Figure 7 micromachines-11-00726-f007:**
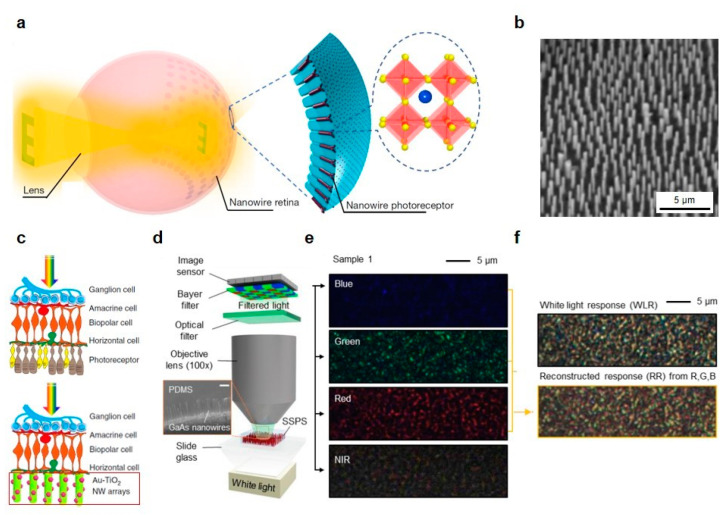
(**a**) Schematic of the spherical biomimetic electrochemical eye (EC-EYE) system with an enlarged illustration of perovskite nanowire photoreceptor arrays. (**b**) SEM image of perovskite nanowires. (**c**) Schematic of the rod and cone cells in necrotic photoreceptor layer replacement with artificial photoreceptors of Au-TiO_2_ nanowire array. (**d**) Schematic of the optical setup for the selective absorption measurement of the selective and sensitive photon sieve (SSPS). SEM image of the SSPS (inset). (**e**) Filtered optical microscope images of SSPS with 450 nm, 550 nm, 632.8 nm and 750 nm filtered lights. (**f**) White light response (top) and reconstructed response (bottom) from red, green and blue images. (**a**,**b**) reprinted by permission from Springer Nature: Springer *Nature* [121], Copyright 2020; (**c**) reprinted by permission from Springer Nature: Springer *Nature Communications* [122], Copyright 2018; (**d**–**f**) reprinted from [63], with permission from Wiley.

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
