# Peer review of "Recent Advances in Vertically Aligned Nanowires for Photonics Applications"

_micromachines, 2020, doi:10.3390/mi11080726_

Round 1
Reviewer 1 Report
Manuscript Title: Recent Advances in Vertically Aligned Nanowires for Photonics Applications
Summary: The manuscript is an comprehensive review of vertically aligned nanowires which can be used for photonics applications. This is an important and timely review article that will be useful to engineers and scientists alike. The following revisions are recommended.
Recommendation: Minor Revisions
Revisions:
- The authors can add a section specifically talked about the different materials used to develop nanowires for photonics applications. A discussion about the importance of the chemical composition of the material used to fabricate nanowires will be helpful.
- The use of nanopatterned bulk metallic glass in photonics and biosensors is also emerging as an alternative to traditional materials. These materials also allow for thermoplastic forming to be used in developing nanowires, which is distinct from lithography techniques described in the paper. This can be added to the review with appropriate citations.
- The applications section of the review article can also be expanded to include potential applications of nanowires photonics in molecular diagnostics, computational technology and alternative energy.
Reviewer 2 Report
The manuscript “Recent Advances in Vertically Aligned Nanowires for Photonic Applications“ by Sehui Chang et al. is a review on applications of vertically aligned nanowires and their production.
This topic is interesting for publication in Micromachines.
The introduction gives a good overview about nanowires in photonics and specializes then on vertically aligned nanowires.
Section 2 deals with the operation principles and introduces leaky/guided waveguide modes.
In section 3 geometrical variations like periodicity and morphology are discussed.
In section 4 several fabrication techniques are introduced and in section 5 example applications are discussed.
The manuscript is well written and easy to follow for a general readership.
Several things could be improved.
1. Title/line 11/line 31: vertically aligned or vertically-aligned
2. One promising application of nanowire arrays would be the generation of hard X-rays by femtosecond laser pulses. The benefit would be the enlarged interaction volume. Samsonova, Z.; Höfer, S.; Hollinger, R.; Kämpfer, T.; Uschmann, I.; Röder, R.; Trefflich, L.; Rosmej, O.; Förster, E.; Ronning, C.; Kartashov, D.; Spielmann, C. Hard X-ray Generation from ZnO Nanowire Targets in a Non-Relativistic Regime of Laser-Solid Interactions. Appl. Sci. 2018, 8, 1728 https://doi.org/10.3390/app8101728
3. One promising experiment for single nanowires would be Strong-field driven Nanolasers. Nano Lett. 2019, 19, 6, 3563–3568, Publication Date:May 22, 2019
https://doi.org/10.1021/acs.nanolett.9b00510
